# Two-tier supramolecular encapsulation of small molecules in a protein cage

Thomas G. W. Edwardson [1], Stephan Tetter [1] & Donald Hilvert [1]✉

Expanding protein design to include other molecular building blocks has the potential to increase structural complexity and practical utility. Nature often employs hybrid systems, such as clathrin-coated vesicles, lipid droplets, and lipoproteins, which combine biopolymers and lipids to transport a broader range of cargo molecules. To recapitulate the structure and function of such composite compartments, we devised a supramolecular strategy that enables porous protein cages to encapsulate poorly water-soluble small molecule cargo through templated formation of a hydrophobic surfactant-based core. These lipoprotein-like complexes protect their cargo from sequestration by serum proteins and enhance the cellular uptake of fluorescent probes and cytotoxic drugs. This design concept could be applied to other protein cages, surfactant mixtures, and cargo molecules to generate unique hybrid architectures and functional capabilities.

[1] Laboratory of Organic Chemistry, ETH Zurich, 8093 Zurich, Switzerland. ✉email: donald.hilvert@org.chem.ethz.ch

Nature uses lipids, polysaccharides, and proteins to create physical barriers for the compartmentalization of molecules and chemical processes. The myriad roles that these structures serve across broad length scales have inspired efforts to generate analogous artificial systems[1–3], both to gain insight into the origin and function of biological infrastructure and to create next-generation nanotechnology. Among biomolecular compartments, protein-based containers are attractive owing to their well-defined structures, diverse functionalities, and genetic manipulability[4]. Natural and artificial protein cages are ideal hosts for macromolecules, a property that has been exploited to produce a wealth of engineered nanosystems[5–11], from catalytic compartments[12,13] to virus mimics[14]. Critical to the success of these designs has been predictable molecular recognition between components. Perhaps, for this reason, the incorporation of biological components beyond proteins and nucleic acids has not been extensively explored, as this requires developing an additional set of, likely harder to predict, self-assembly rules. Considering the structural and functional complexity that nature achieves through the coordinated self-assembly of diverse (macro)molecular components, expansion of protein cage engineering to include other molecular species has considerable potential to advance structural complexity and functional scope[15–18].

From the hydrophobic pockets of globular proteins[19] to the lipidic cores of lipoproteins[20] and lipid droplets[21], confined nonpolar environments are commonly found in biology as a solution to traffic small molecules. In lipoproteins, such compartmentalization is achieved by combining the amphiphilic self-assembly of lipids with proteins that stabilize the complex and act as biorecognizable barcodes, guiding their cargo to a suitable address (Fig. 1a). Inspired by these assemblages, we sought to integrate hydrophobic compartmentalization and engineered protein cages, with the aim of creating a general and modular platform that could be adjusted for the carriage of nonpolar guest molecules.

In our two-tier host–guest approach, designed protein containers possessing highly positively charged interiors are used to nucleate anionic surfactant molecules into micellar aggregates within their lumenal cavities at concentrations well below the critical aggregation concentration. The resulting protein-scaffolded micelles are stable with well-defined stoichiometry, and efficiently encapsulate nonpolar small molecules within their hydrophobic cores. Moreover, altering the lipid composition allows fine-tuning of the binding affinity and release kinetics for different cargo. This hybrid system protects cargo from sequestration by serum proteins and increases cellular uptake, and therefore biological activity, of small molecule therapeutics. Our findings demonstrate that a beneficial combination of electrostatically driven and amphiphilic self-assembly within stable protein capsules provides a means to access composite compartments with practical function.

## Results

**Protein cage-templated lipid assembly.** As a protein scaffold, we chose OP[22], a small porous capsid based on the computationally designed O3-33[23], which has a positively charged interior cavity provided by 144 arginine residues presented on its lumenal surface. After expression in *Escherichia coli*, the OP protein is isolated as a complete octahedral assembly comprising 24 monomers with a ~3.5 nm pore on each of its six faces (Fig. 1b). The external diameter of the cage is ~13 nm and the diameter of the spherical internal cavity is ~8 nm. This highly stable, porous structure has overall dimensions that are well suited to scaffold small micellar aggregates within its interior.

Sequestration of small molecules using the OP cage relies on creating a favorable environment for nonpolar species within the lumen by hierarchical supramolecular assembly. Combining orthogonal self-assembly modes within an individual nanostructure has been successful in DNA nanotechnology, providing access to unique structures and functions[24,25], and we wondered whether this approach could be applied to a protein scaffold. The positively charged OP capsid allows the use of charge complementarity to drive the internalization of negatively charged amphiphiles, which in turn phase separate to maximize both guest–guest and host–guest interactions, creating a hydrophobic core (Fig. 1c). The protein-scaffolded lipid droplet can then sequester nonpolar small molecules, in a manner akin to natural

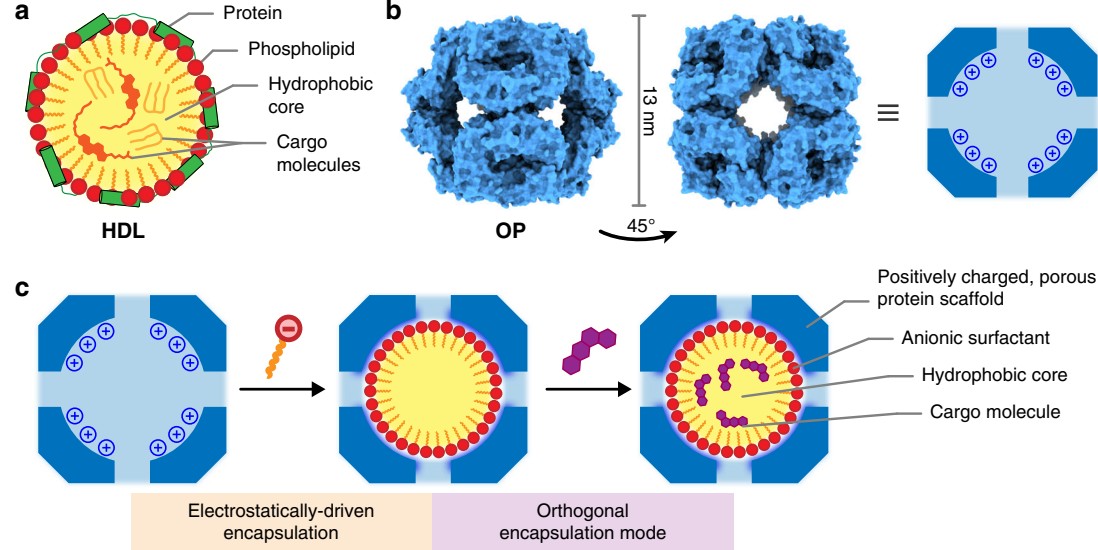

**Fig. 1 Self-assembly of lipoprotein-mimetic capsids. a** Cartoon depiction of a high-density lipoprotein (HDL) particle, showing the charged phospholipids, proteins, and hydrophobic cargo molecules. **b** Surface representations of the OP protein cage viewed along the twofold (left) and fourfold (right) symmetry axes and a cartoon cut-away depicting the positively charged interior. **c** Two-tier encapsulation concept: electrostatic attraction drives the encapsulation of anionic surfactants, which phase separate due to their high effective concentration, forming micellar aggregates within OP cages. The hydrocarbon core of these stable protein–surfactant complexes then sequesters nonpolar small molecules by means of the hydrophobic effect.

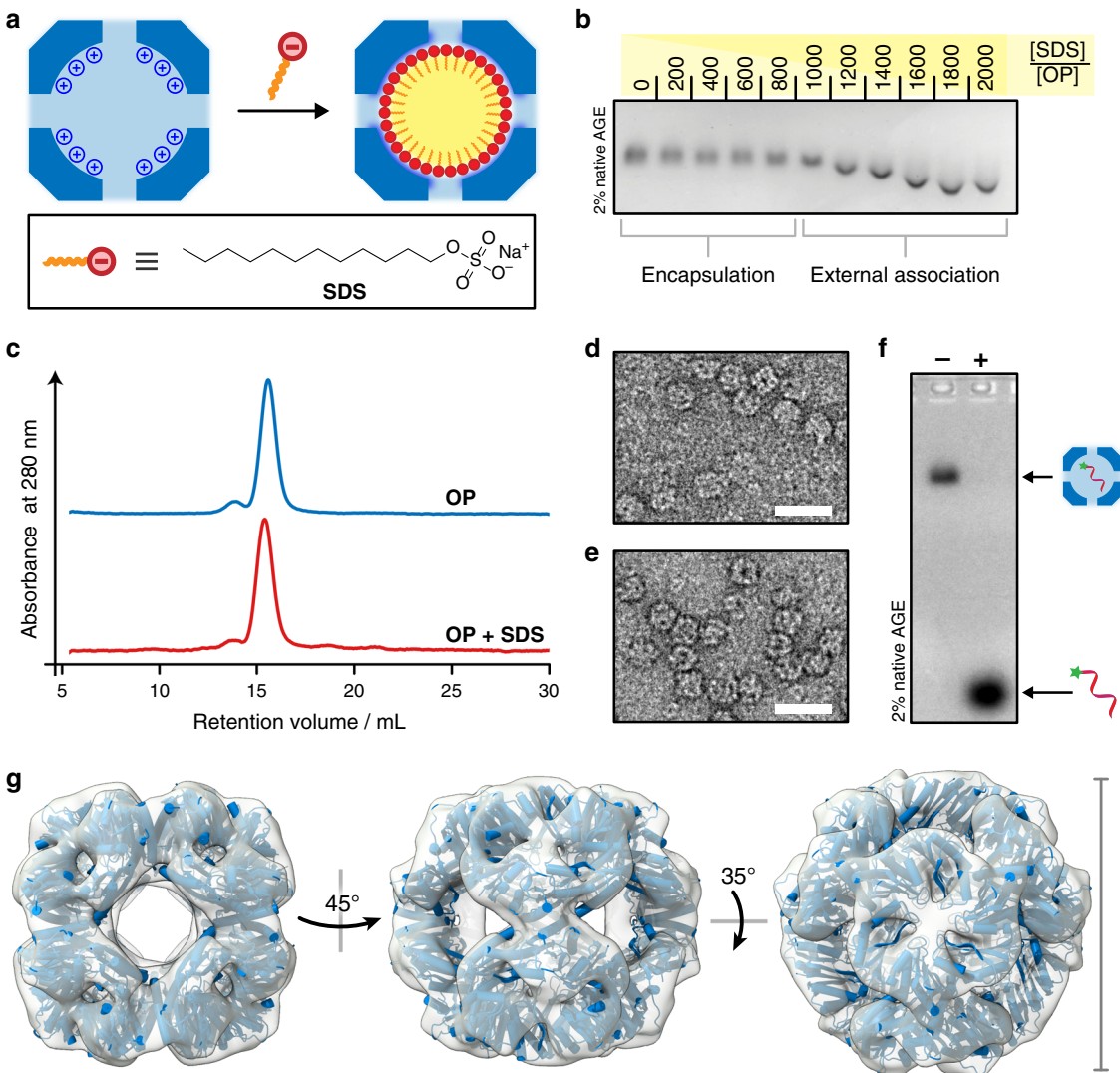

**Fig. 2 Capsid-templated micelle formation. a** Cartoon depiction of protein cage-templated micellization of sodium dodecyl sulfate (SDS) molecules. **b** Native gel electrophoresis of the OP cage in the presence of increasing molar equivalents of SDS. **c** Size-exclusion chromatography of the OP cage before and after incubation with 800 equivalents of SDS. **d** Transmission electron micrograph of empty OP cages and **e** OP cages in the presence of SDS (800 equiv.); scale bars are 30 nm. **f** Encapsulation of fluorophore-labeled ssDNA visualized by native gel electrophoresis. In the absence of SDS (−) the OP cage internalizes the ssDNA probe quantitatively. After internalization of the SDS molecules (+) the OP cage can no longer encapsulate the oligonucleotide probe. **g** 3D volumes determined by cryo-EM for OP:SDS (transparent surface) overlaid onto the crystal structure (blue ribbons and cylinders, PDB: 6FDB) show that the protein cage structure is retained. Scale bar = 13 nm. Source data are provided as a Source Data file.

lipoproteins, albeit through an alternative, non-natural architecture templated by the porous OP cage.

Sodium dodecyl sulfate (SDS) was initially chosen as the anionic surfactant (Fig. 2a) due to its aqueous solubility and the potential for favorable electrostatic interactions and hydrogen bond formation between the sulfate headgroup and the many arginine residues on the interior surface of the OP capsid[26]. As SDS is typically used as a protein denaturant, it was important to establish that charge and shape complementarity of the OP capsid could be leveraged to drive the formation of a scaffolded micelle, rather than allow SDS to disrupt the protein structure. Based on the volumetric capacity of the OP cavity (~256 nm³) and the average volume occupied by an SDS molecule in a micelle (Supplementary, Methods section), we expected that ~730 molecules could be encapsulated within a single OP capsid. As such, OP was treated with increasing molar equivalents of SDS, up to 2000 molecules per capsid, and analyzed by native gel electrophoresis (Fig. 2b). Indeed, no change in band mobility was observed up to ~800 equivalents of SDS, consistent

with internalization, while more than 1000 equivalents gave rise to bands with slightly increased mobility. As dynamic light scattering (Supplementary Fig. 1a) revealed no significant change in particle diameter, this change can be attributed to an increase in surface charge, indicating exposure of some sulfate groups. Denaturation of the protein structure required ~100-fold higher SDS concentrations (Supplementary Fig. 1c). The quaternary structure of OP was further interrogated in the presence of 800 equivalents of SDS per capsid, at molarities below the critical micelle concentration (CMC) of the surfactant. Zeta potential measurements corroborated the native gel analysis, showing a negligible change in surface charge (Supplementary Fig. 1d) and size-exclusion chromatography, transmission electron microscopy and cryo-electron microscopy (Fig. 2 and Supplementary Figs. 2–5) confirmed that there was no change to protein cage structure.

Localization of SDS molecules in the lumen via the intended sulfate-guanidinium interactions should block competing negatively charged guests from entry. This obstruction would be due

to both negation of the high positive charge on the cage, which is the driving force for encapsulation, and occlusion of the entry pores. As oligonucleotides are internalized by empty OP capsids rapidly and with high affinity[22], they are an ideal probe to test this hypothesis. Fluorophore-labeled 21 nt ssDNA was added to either empty OP cages or those pre-incubated with SDS, and the complexes were analyzed by native gel electrophoresis (Fig. 2f). While the empty OP cages encapsulate the probe quantitatively, the OP:SDS complexes are unable to encapsulate the DNA strands, consistent with SDS molecules occupying the lumenal cavity.

**Small molecule encapsulation**. Deeper characterization of the internal structure of the protein-lipid complex was achieved using the solvatochromic dye Nile Red. This hydrophobic small-molecule fluorophore is poorly emissive in aqueous media but exhibits strong fluorescence in nonpolar environments[27]. To discern if the OP cage chaperoned SDS molecules into hydrophobic aggregates, Nile Red fluorescence was measured in the presence of each component of the system (Fig. 3a). Below its CMC (4–5 mM in PBS buffer)[28], SDS had a negligible effect on the fluorescence of a 500 nM aqueous solution of Nile Red. Likewise, the OP cage did not enhance the emission of the fluorescent probe. However, in the presence of both OP and 800 equivalents of SDS, a large increase in Nile Red fluorescence was observed. This increase in signal was accompanied by a blue-shift in the emission maximum, indicating localization of the probe within the lipidic core of the capsid[27]. This fluorescence increase

was also monitored over time at different NaCl concentrations, revealing assembly kinetics that are inversely dependent on ionic strength, consistent with electrostatically driven complex formation (Supplementary Fig. 6). Although various nanoparticles have been coated with peptides and proteins[29–32], affording a range of functional hybrid assemblies[33], the use of a well-defined, stable protein shell to template lipid assembly offers an alternative means of creating hybrid containers, simplifying formulation, and allowing surfactant composition to be tailored to specific applications.

With the presence of a hydrophobic core established, Nile Red was employed to verify the number of SDS molecules encapsulated in each OP cage. Fluorescence monitoring of two equivalents of Nile Red in the presence of OP:SDS complexes with increasing SDS content revealed a plateau at around 800 surfactant molecules per cage (Fig. 3b, c), corroborating the native gel analysis (Fig. 2b). Although the exact internal organization of the SDS aggregates is unknown, the packing density suggests that the molecules are arranged with some structural similarity to their typical oblate ellipsoid micellar form[34].

For applications in molecular transport, cargo loading capacity is an important parameter. We therefore used fluorescence titration to determine the number of Nile Red molecules that could be accommodated per protein–micelle complex. Single equivalents of Nile Red were added stepwise to a solution of OP:SDS complexes and a fluorescence spectrum was measured after each addition (Fig. 3d). For 1–5 molecules of Nile Red per capsid,

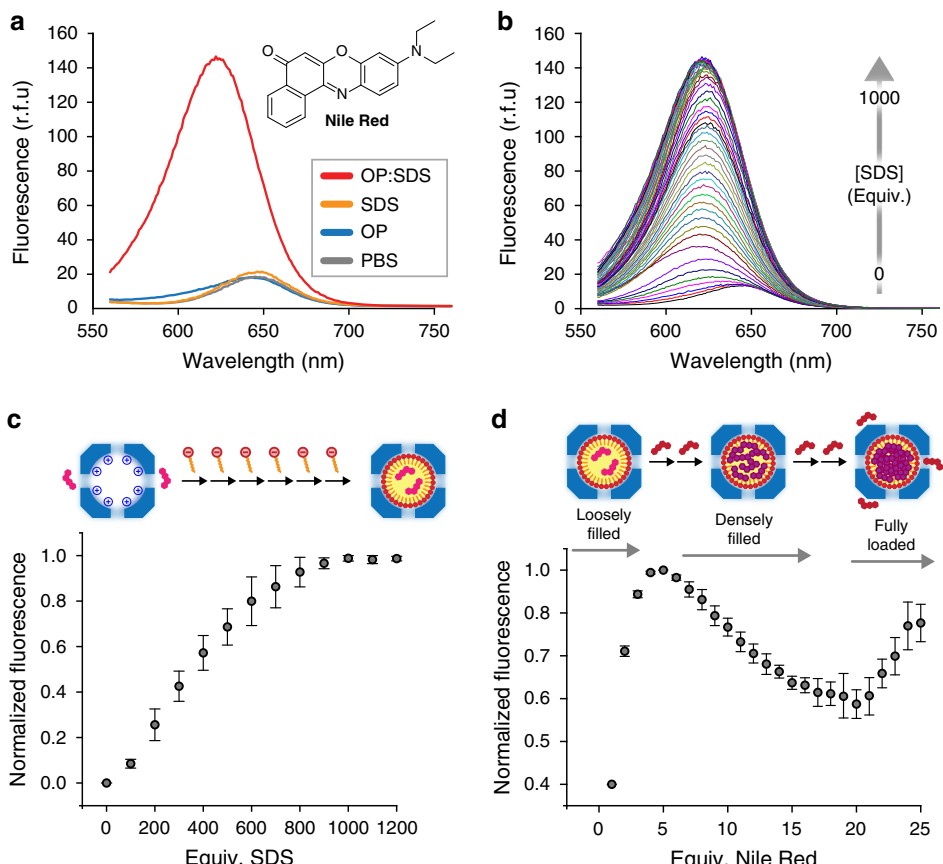

**Fig. 3 Hydrophobic core formation and cargo capacity. a** Steady-state fluorescence emission spectra of Nile Red in the presence of OP, SDS, and the OP:SDS complex, with reference to PBS buffer. **b** Steady-state fluorescence emission spectra of Nile Red in the presence of OP with increasing molar equivalents of SDS. **c** Normalized fluorescence emission of Nile Red at 620 nm vs. molar equivalent of SDS with respect to OP capsids ($n = 3$, error bars: standard deviation). **d** Normalized fluorescence emission at 620 nm vs. molar equivalents of Nile Red added to OP:SDS complexes containing 800 equiv. of surfactant per capsid ($n = 3$, error bars: standard deviation). Source data are provided as a Source Data file.

an increase in fluorescence emission is observed upon the addition of each equivalent. Fluorescence then decreases steadily from 5 to 20 molecules of Nile Red and increases again above 20 equivalents. In the first stage, the addition of each additional Nile Red molecule gives a diminishing increase in emission due to self-quenching between the dyes[35], a process that begins to dominate in the second stage owing to the high effective molarity of the fluorophore. In the third stage, the emission increase per equivalent is very close to that observed for Nile Red in bulk media and is associated with a redshift (Supplementary Fig. 7), indicating localization in an aqueous environment. From these data, it can be concluded that the loading capacity of the OP:SDS complex is around 20 molecules per capsid. The effective concentration of the fluorophore (~130 mM) in fully loaded cages is two orders of magnitude higher than the solubility limit for Nile Red in water (0.6 mM).

**Cellular uptake.** Encouraged by our recent finding that OP can deliver short interfering RNA to the cytosol of mammalian cells and induce efficient gene knockdown[22], we assayed the micelle-containing OP cages for their ability to improve cellular uptake of poorly soluble compounds. Human cervical cancer cells (HeLa) were treated with either OP:SDS complexes carrying Nile Red or the free fluorophore itself. Analysis by flow cytometry (Fig. 4a and Supplementary Fig. 8) showed that OP:SDS complexes enhance the cellular uptake of the small-molecule fluorophore. This finding was confirmed by confocal fluorescence microscopy (Fig. 4b and Supplementary Fig. 9), which also revealed that Nile Red was distributed throughout the cell. Based on the

intracellular trafficking of OP cages, the majority of which localize in endosomes[22], this result suggests that the Nile Red cargo escapes from the capsids after internalization and shuttles to hydrophobic environments in the cell. To test this hypothesis, Atto425-labeled OP capsids were used (Supplementary Fig. 9). Although Nile Red diffuses within the cell, the OP signal is clustered in punctate foci, confirming that the majority of OP capsids are retained in the endolysosomal system, while the small molecule cargo is released to the cytoplasm.

**Tuning lipid composition.** Plasma lipoprotein cores comprise an assortment of hydrophobic molecules, which combine to enhance particle stability and cargo molecule solubilization. It follows that altering the surfactant composition within OP cages should provide a means to tune their physical properties. We chose the endogenous steroid cholesterol sulfate (CS) as an additional component since cholesterol is an important constituent of lipoproteins and is known to stabilize membranes. A surfactant composition of 75% SDS and 25% CS was found to be well tolerated by the OP capsid, affording encapsulation complexes with Nile Red (Supplementary Figs. 10 and 11). Cryo-EM analysis of these OP:SDS:CS complexes showed that protein structure was unperturbed and also revealed electron density within the interior cavity (Supplementary Fig. 12). Furthermore, flow cytometry of cells treated with Nile Red-loaded OP:SDS:CS complexes revealed a 2.5-fold enhancement in intracellular delivery compared to OP:SDS (Fig. 4d). This result is likely due to the increased stability of the encapsulation complex provided by the addition of CS, which is evident from a comparison of the Nile Red release rate from

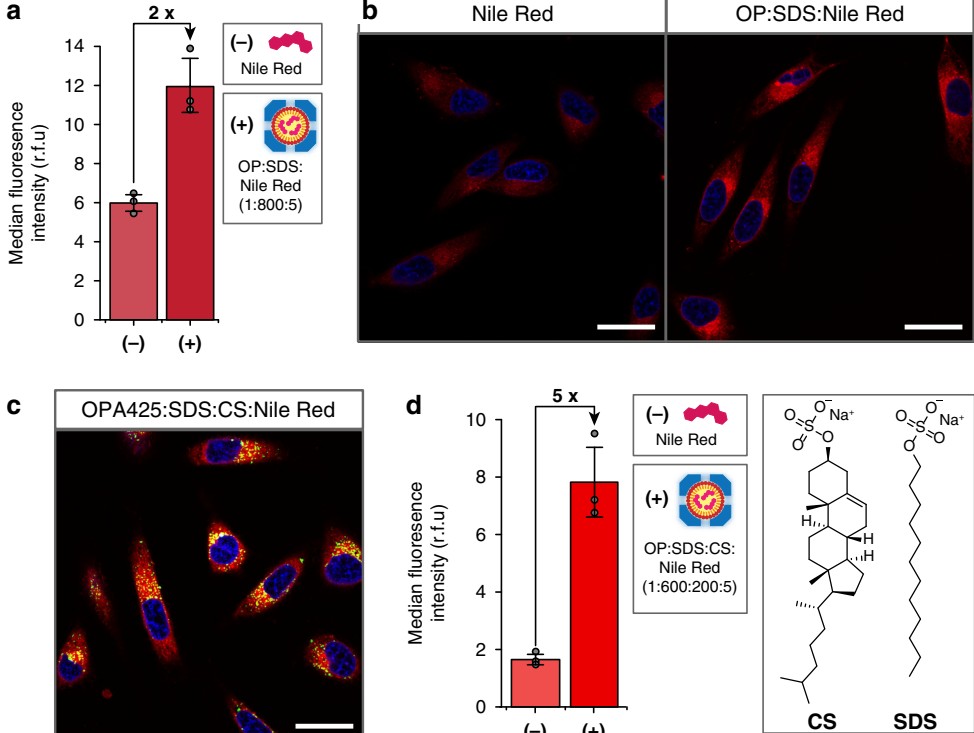

**Fig. 4 Intracellular delivery of fluorescent cargo. a** Flow cytometry dataset for HeLa cells treated with either free Nile Red or Nile Red packaged in OP:SDS complexes, with five Nile Red molecules per capsid (*n* = 3, error bars: standard deviation). **b** Confocal fluorescence microscopy of HeLa cells treated with Nile Red as free molecules (left panel) or packaged in OP:SDS complexes (right panel). Blue: Hoechst 33342 (nucleus), red: Nile Red, scale bars are 30 μm. **c** Confocal fluorescence microscopy of HeLa cells treated with OP:SDS:CS:Nile Red complexes where OP capsids are labeled with Atto425. Blue: Hoechst 33342 (nucleus), red: Nile Red, green: Atto425 (OP), scale bar is 30 μm. The individual channels can be found in Supplementary Fig. 8. **d** Flow cytometry comparison of HeLa cells treated with free Nile Red (−) or Nile Red packaged in OP:SDS:CS complexes (+), where the ratio of OP:SDS:CS is 1:600:200 and there are 5 molecules of Nile Red per capsid (*n* = 3, error bars: standard deviation). Source data are provided as a Source Data file.

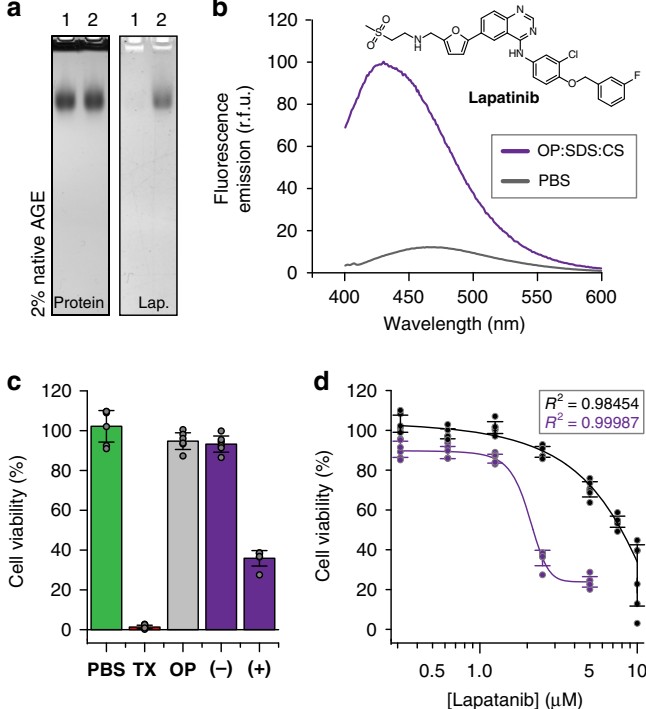

**Fig. 5 Stable encapsulation and improved potency of lapatinib. a** Native agarose gel of OP:SDS:CS complexes visualized with Coomassie blue for protein (left) and by lapatinib fluorescence (right). Lane 1—OP:SDS:CS (1:600:200); Lane 2—OP:SDS:CS:lapatinib (1:600:200:10). **b** In the presence of OP:SDS:CS complexes lapatinib fluorescence increases significantly and the emission maximum shifts to lower wavelengths, consistent with encapsulation in a nonpolar environment. **c** Viability of HeLa cells after treatment with 2.5 μM of free lapatinib (−) or lapatinib encapsulated in OP:SDS:CS complexes (+). Control samples are PBS buffer, Triton X-100 (TX), and OP:SDS:CS complexes without the encapsulated drug (OP) (n = 6, error bars: standard deviation). **d** Dose–response comparison of cell viability after treatment with free lapatinib (black), or lapatinib encapsulated in OP:SDS:CS complexes (purple) (n = 6, error bars: standard deviation). For each concentration, a loading of 10 lapatinib molecules per capsid was used. Source data are provided as a Source Data file.

OP:SDS and OP:SDS:CS complexes (Supplementary Fig. 13). Nevertheless, cargo is still efficiently released from the capsid after cellular uptake (Fig. 4c and Supplementary Fig. 9). The effect of CS addition highlights the inherent modularity of this system and the potential for tuning the surfactant composition to obtain desired physical properties for different cargo molecules.

**Delivery of bioactive molecules**. To evaluate the encapsulation, transport, and intracellular release of an active pharmaceutical compound, we chose the dual tyrosine kinase inhibitor lapatinib[36]. This drug is used as a therapy for solid tumors and has been shown to benefit from nanoparticle-mediated delivery due to its poor solubility and serum protein binding[37]. Lapatinib is encapsulated by OP:SDS:CS complexes with high affinity (Fig. 5a, b). These protein–surfactant assemblies not only retain the drug after 72-h dialysis against medium containing bovine serum albumin (BSA) (Supplementary Fig. 14), but, strikingly, can even extract it from pre-formed BSA–lapatinib complexes (Supplementary Fig. 15).

To test how these stable encapsulation complexes perform, we assessed effective cytotoxicity of lapatinib-loaded OP:SDS:CS on HeLa cells. After 18 h treatment with either free lapatinib or

lapatinib packaged in OP:SDS:CS cages, cell viability was monitored (Fig. 5c). At a concentration of 2.5 μM, free lapatinib had a negligible effect on cell viability. However, at the same concentration, with lapatinib packaged at a ratio of 10 molecules per OP:SDS:CS, potency was notably enhanced, killing 60% of the cells. Importantly, the surfactant-filled OP cages themselves showed negligible toxicity, confirming that the increased efficacy was due to the delivery of lapatinib to its intracellular target. A dose–response comparison of free and encapsulated drug (Fig. 5d), reveals that OP:SDS:CS complexes provide a 3.5-fold decrease in IC50, an effect that is better than other nanoparticle-based systems for lapatinib delivery[37]. This improvement can likely be further boosted by appending ligands that target specific cell surface receptors to the exterior of the OP capsid. Furthermore, the ability of OP to penetrate diverse cell types[22] suggests that this delivery system could have broad applicability.

## Discussion

The integration of amphiphilic self-assembly with designed protein cages affords a system reminiscent of serum lipoproteins, but with a distinct supramolecular architecture. The unique structure of the porous OP protein cage serves as a stable template to guide the assembly of surfactant aggregates and allow the influx and efflux of guest molecules. Extension of this general strategy to other natural and artificial protein scaffolds has the potential to generate libraries of hybrid systems with their own distinct properties.

This approach can be compared to the use of constrained chemical[38] or enzymatic[39,40] polymerization within protein cages to create hydrophobic cores and provide reactive handles for covalent attachment of drug and dye molecules[34]. Our bio-inspired strategy offers a non-covalent alternative to such systems. By avoiding chemical transformations that produce highly stable particles, these supramolecular complexes more closely mimic the dynamic, reversible properties of biological assemblies. As such, the resulting hybrid particles are ideally suited to bind, transport, and deliver nonpolar cargo molecules to cells, in a manner analogous to lipoproteins[16]. Importantly, as demonstrated with lapatinib, a practical balance of serum stability, efficient cellular uptake, and intracellular release can be obtained by suitably modulating the internal lipid composition.

Since capsid templation removes the need for pre-formation of stable lipid nanoparticles, a wide range of amphiphilic molecules can be employed, as long as the mixture possesses sufficient complementary charge to drive encapsulation. Furthermore, exploiting the combination of electrostatic cage–cargo interactions and hydrophobic cargo–cargo interactions, alternative lipidic structures, such as liposomes, or altogether different amphiphiles, such as block copolymers, could be employed to create hybrid core–shell assemblies from larger protein cages. It stands to reason that such systems could be used to encapsulate all manner of nonpolar molecules, with careful tuning of the surfactant formulation enabling optimization of the loading capacity, binding affinity, and release kinetics for each specific cargo.

De novo design has provided access to an array of novel protein assemblies by traversing unexplored sequence, structure, and fitness landscapes[41,42]. Introducing other molecular building blocks to designed assemblies increases the dimensionality of this parameter space exponentially[17,18,43–45]. The emergent properties that result can take us one step closer to matching the structural sophistication and functional elegance that we see in nature.

## Methods

**Materials**. All chemicals were used as supplied without further purification. Iso-propyl-β-D-thiogalactopyranoside (IPTG) was purchased from Fluorochem (UK).

Lysozyme was purchased from PanReac Axon Lab AG (Switzerland). For His-tagged protein isolation, Ni-NTA Agarose from Qiagen (Germany) was used. DNase I was from Roche (Switzerland) and RNase A was from Merck (Germany). Sodium dodecyl sulfate and cholesterol sulfate were purchased from Sigma-Aldrich (Merck, Germany). Oligonucleotides were purchased from Microsynth AG (Switzerland). All cell culture media and reagents were from Gibco (ThermoFisher Scientific Inc., USA).

**Instrumentation**. Protein quantification was carried out using a NanoDrop 2000c spectrophotometer from ThermoFisher Scientific Inc. (USA). All size-exclusion chromatography was carried out on an NGC$^{TM}$ Medium-Pressure Chromatography System from Bio-Rad Laboratories, Inc (USA). Agarose gel electrophoresis (AGE) was performed on Mini-Sub® cell GT from Bio-Rad Laboratories, Inc. (USA). Gel images were captured using an EOS 1100D from Canon (Japan). Electron microscopy images were obtained on a TFS Tecnai F20 FEG (USA). Fluorimetry was carried out on a QuantaMaster$^{TM}$ 50 fluorometer from Photon Technology International (USA). Confocal fluorescence microscopy images were obtained on an SP8-AOBS from Leica (Germany). Flow cytometry was carried out on an LSRFortessa from BD Biosciences (USA).

**Software**. The following software was used for data collection: DSLR Remote Pro v1.4 (Fluorescence Gel images), Epson SilverFast v8.0.1r16 (Coomassie Gel images), Biorad ChromLab v4.0.0.25 Std Edition (Size-exclusion Chromatography), FEI EPU Software v1.12 (Electron microscopy), Felix X32 PTI software v1.2 Build 56 (Fluorimetry), BD FACSDiva 8.0.1 (Flow cytometry), Leica Application Suite X v3.7.0.20979 (Confocal Microscopy), Molecular Devices SoftMax Pro v5.4 (Microplate reader). For data analysis, the following software was used: Leica Application Suite X v3.7.0.20979 (Confocal Microscopy), Adobe Photoshop 19.1.5 (Gel images), FlowJo v10.1 (Flow cytometry data), Microsoft Excel—MS Office Professional Plus 2013 (Fluorimetry, flow cytometry, size-exclusion chromatography, cytotoxicity), OriginPro b9.3.1.273 (Fluorimetry, flow cytometry, cytotoxicity), UCSF Chimera v1.10.2 Build 40686 (Protein structure), UCSF ChimeraX v0.91 (Protein structure), Relion v3.0 (cryo-electron microscopy), Fiji 1.51n (Image processing).

**Protein production**. Proteins were expressed in *E. coli* strain BL21-Gold(DE3). Cells were grown at 37 °C in LB medium containing kanamycin sulfate (86 μM) until OD$_{600}$ reached 0.6–0.8, and protein over-expression was induced with Iso-propyl β-D-1-thiogalactopyranoside (0.1 mM). After culturing for ~18 h at 25 °C, cells were harvested by centrifugation (5000 × *g*) at 4 °C for 15 min. Cell pellets were stored at −20 °C until purification. OP capsids were isolated from *E. coli* cell pellets and purified by Ni-affinity and size-exclusion chromatography as previously reported[22].

**Preparation of OP:SDS and OP:SDS:CS complexes**. Protein cage-micelle complexes were formed directly from purified, empty OP cages, and aqueous solutions of anionic surfactants. Unless otherwise specified, the molar ratio of total surfactant molecules to OP capsids is 800:1 in all experiments. Buffers used were PBS (9.5 mM Na$_2$HPO$_4$, 1.4 mM KH$_2$PO$_4$, 136 mM NaCl, 2.7 mM KCl, pH 7.4) and TSEC (25 mM Tris-HCl, 200 mM NaCl, 5 mM EDTA, pH 7.4). To form complexes, appropriate volumes of concentrated SDS solution (1–100 mM in PBS) or CS solution (8–16 mM, DMSO) were first diluted in PBS buffer to concentrations below 1 mM to avoid protein denaturation. Then, the necessary volume of OP solution (2–20 μM capsid, PBS, or TSEC buffer) was added and the mixture was incubated for at least 1 h at room temperature to allow complete complex formation. For small-molecule encapsulation, concentrated solutions of fluorescent probe/drug in acetone or DMSO were added to the pre-formed OP:SDS:CS complexes and incubated for a further 15 min at room temperature. In each case, the total fraction of organic solvent was kept below 10% v/v.

**Native agarose gel electrophoresis**. All native gel electrophoresis was carried out using 2% (w/v) agarose gels in Tris-acetate-EDTA buffer (40 mM Tris-HCl, 19 mM acetic acid, 1 mM EDTA, pH 8.3). After the visualization of fluorescent molecules by UV transillumination, gels were stained with Coomassie blue for protein visualization. In a typical experiment, ca. 100 pmol of capsid (with respect to monomer) was loaded per lane in 10 μL of buffer with an additional 2 μL of 70% (v/v) aqueous glycerol for loading. The sequence of the HPLC-purified Atto488-labeled DNA used is Atto488-TTAATTAAAGACTTCAAGCGG. This oligonucleotide is encapsulated quantitatively by the OP cage at the concentrations used and the fluorophore is partially quenched upon encapsulation[22].

**Size-exclusion chromatography**. Analytical SEC was carried out on a Superose 6 Increase 10/300 GL column (GE Healthcare, USA). Samples were 800 μL of 10–50 μM protein monomer and the mobile phase was 0.75× TSEC buffer. Peaks were detected by absorbance at 280 nm.

**Dynamic light scattering and zeta potential**. Measurements were carried out on Zetasizer Nano (Malvern Instruments, UK) at 25 °C. For DLS, samples prepared from 0.22 μm filtered solutions of protein and surfactants in 0.75× TSEC buffer were used at concentrations of 12–100 μM of protein monomer. A measurement angle of 173° was used for all samples. For zeta potential measurements, the OP protein (75–100 μM monomer, TSEC buffer) was dialyzed against deionized water using 7000 Da MWCO Slide-A-Lyzer cassettes (ThermoFisher Scientific, USA). The protein was then filtered, quantified, and diluted to 48 μM monomer in water with or without 800 equivalents of SDS. Before measurement, the sample pH was adjusted to 7.5 using aqueous NaOH.

**Transmission electron microscopy**. Negatively stained transmission electron microscopy (TEM) was carried out as reported previously[46]. For all TEM experiments, samples were between 2 and 4 μM of OP monomer in PBS buffer.

**Nile Red fluorescence**. For a typical fluorimetry experiment, 800 μL samples in PBS buffer were used. Stock solutions of Nile Red in acetone:water (1:1 v/v), or DMSO at concentrations of 50–500 μM were used to obtain final concentrations in the nanomolar range. SDS concentrations were below the critical micelle concentration (4–5 mM in PBS)[28]. The excitation wavelength was set to 535 nm for all experiments. Replicates are from distinct samples.

**Cryo-EM sample preparation**. Protein cage-micelle complexes were prepared in TSEC buffer and incubated for 2–16 h at room temperature before EM sample preparation. Copper-supported holey carbon grids (Quantifoil, R2/2 Cu 400) were negatively glow discharged at 15 mA for 15 s using a Pelco easiGlow Glow Discharge Cleaning System. Of each sample, 3.5 μL was applied to the grid and blotted with a Vitrobot Mark IV (FEI) for 12–14 s at 25 blot strength and 100% humidity before plunging into liquid ethane.

**Cryo-EM data acquisition**. Grids were imaged on an F20 microscope (FEI) equipped with a Falcon II direct electron detector (FEI) at 200 kV. Images were collected at a dose of 30–35 $e^-$ Å$^{-2}$, a magnification of 62,000-fold (1.8 Å pixel size), and a defocus range from −1.8 to −3.3 μm.

**Cryo-EM data processing**. Data were processed in RELION-3.0[47]. CTF estimation was performed with GCTF[48]. Bad images were excluded based on metadata and manual inspection. Laplacian-of-Gaussian-based autopicking was used on a subset of images to generate 2D templates for autopicking on the entire dataset. Particles were then 2D-classified over several rounds. Initial models were generated imposing octahedral symmetry. Both 2D templates and 3D models were generated independently for each dataset to avoid bias. Particles from 2D classification were further classified in 3D, and the best 3D class refined and post-processed in RELION-3.0. Rigid-body fitting of the previously crystallized OP cage structure was performed using ChimeraX[49].

**Effective concentration calculations**. The volume of the OP lumenal cavity (256 nm$^3$) was estimated as a sphere with a radius of 3.94 nm. This radius was determined by averaging the distances between lumenally exposed residues from the reported crystal structure[22], using the UCSF Chimera software[50]. The effective concentration of SDS was estimated as the number of moles SDS per lumenal cavity volume. The expected number of SDS molecules was estimated by first determining the volume occupied by an individual SDS molecule in a micelle, from the reported average values of SDS micelle radii (1.75 nm) and aggregation number ($n = 64$)[51]. Division of the OP cavity volume by the average volume of a single SDS molecule packed into micellar aggregates gives a value of 729 molecules per OP cage. This estimate is within the error of the experimental results, which suggest that the cage accommodates ~800 molecules.

**Cell culture**. HeLa cells were maintained in Dulbecco's Modified Eagle Medium (high glucose) supplemented with 10% fetal bovine serum (FBS), 2 mM L-gluta-mine, 2 mM GlutaMAX and 1 μg mL$^{-1}$ gentamicin. Cells were cultured in 5% CO$_2$ at 37 °C and typically split in a 1:4 ratio every 3 days. Passage numbers between 7 and 20 were used for all experiments.

**Flow cytometry**. HeLa cells were seeded at a density of 30,000 cells per well in a 24-well plate in 500 μL of culture medium and allowed to recover at 37 °C and 5% CO$_2$ for 24 h to reach 60–80% confluency. Both OP protein and surfactant solutions were sterilized by filtration through a 0.22 μm membrane, and stocks were prepared in sterile PBS. Nile Red solution (50 μM) in 1:1 EtOH:H$_2$O was used without sterilization. OP capsid-micelle complexes were prepared as described above, using component ratios within loading capacity limits to avoid additional purification. For each well, 20 μL of the sample in PBS was added to 200 μL of culture media to give a final concentration of 200 nM. Cells were incubated for 16–20 h in 5% CO$_2$ at 37 °C before washing with PBS and trypsinization (0.05% trypsin-EDTA (ThermoFisher Scientific, USA), 4 min at 37 °C). Cells were collected in cold culture medium and washed twice with cold PBS before resuspension in flow cytometry buffer (PBS with 5% FBS). A representative cytometry analysis with all gating is shown in Supplementary Figure 8. In each experiment the median

fluorescence intensity for cells treated with OP:SDS or OP:SDS:CS complexes is compared to free Nile Red, as the absolute values differ slightly due to instrument settings. Replicates correspond to distinct samples.

**Fluorescent labeling of OP capsids**. To provide a specific handle for fluorophore conjugation, a single serine to cysteine mutation was introduced at residue 38 of the OP protein. This lumenally presented residue was chosen to avoid interfering with the exterior surface of the OP cage, which could disrupt the cellular uptake profile. The gene for this OPS38C variant was generated through "QuikChange" (Agilent) site-directed mutagenesis. The primers used were: OP_S38Cfw; ATCTGCTGGTGAGC AAAACCATTTGCCGTGGTAAAT and OP_S38Crv; ATTTACCACGGCAAAT GGTTTTGCTCACCAGCAGAT. Successful molecular cloning was confirmed by Sanger sequencing (Microsynth AG, Switzerland) of the pET29b(+)_OPS38C plasmid used for protein expression and the protein was produced as previously reported[22]. Labeling of the OP cage with Atto425-maleimide (Sigma-Aldrich) was carried out by mixing purified protein in TSEC buffer with dye solution (10 mM, DMSO) and incubating in the dark overnight at room temperature. To terminate the reaction, 2 equiv. (w.r.t. maleimide) of β-mercaptoethanol were added and after 30-min incubation, the protein was purified using a PD minitrap G-10 column (GE Healthcare, USA). The labeling efficiency was determined from UV-Vis absorbance measurement and the $\varepsilon_{280}$ and $\varepsilon_{439}$ values of Atto425 and the OP protein. For the experiments shown in Fig. 3c and Supplementary Fig. 9, samples with average labeling of 1.9 dyes per OP capsid were used.

**Confocal microscopy**. Cells were seeded at a density of 15,000–20,000 cells per well in a μ-slide 8-well chambered coverslip with ibiTreat surface from ibidi GmbH (Germany) and incubated in 200 μL of culture medium at 37 °C and 5% $CO_2$ for 24 h before sample addition. Sterile samples were prepared in PBS and for each well 10 μL of sample solution was added to 100 μL of culture media to give the desired final concentrations of protein and fluorophore. In all images shown, a loading of five Nile Red molecules per capsid was used, which forms a stable complex and could be applied to cells without purification. Cells were incubated with samples for 24 h in 5% $CO_2$ at 37 °C before washing with PBS and nuclear staining with 100 μL of Hoechst 33342 solution (5 μg mL$^{-1}$ in PBS) at 37 °C for 10–15 min. Cells were then washed twice with PBS and microscopy was carried out at 37 °C in PBS containing 10% FBS. Images were obtained using a 63x objective and excitation/ emission wavelengths were as follows: Hoechst, Ex. 405 nm, Em. 426–472 nm; Atto425, Ex. 458 nm, Em. 499–559 nm and Nile Red, Ex. 514 nm, Em. 580–680 nm.

**Release of Nile Red from protein–micelle complexes**. OP:SDS:NileRed (1:800:2, input ratio) and OP:SDS:CS:NileRed (1:600:200:2, input ratio) complexes were prepared at final concentrations of 0.5 μM OP in PBS buffer. To 200 μL of this solution were added 600 μL of hexane in a screwtop centrifuge tube and the samples incubated at room temperature for 3 or 16 h. The phases were then carefully separated by pipette and multiple fluorescence spectra measured for each sample. For the organic phase, 50 μL sample was diluted with 750 μL hexane and the samples excited at 470 nm and emission monitored from 500 to 700 nm. For the aqueous phase, 40 μL of the sample was diluted with 760 μL of PBS; the sample was excited at 535 nm and emission monitored from 560 to 760 nm. To obtain the plot shown in Supplementary Fig. 13c, the maximum fluorescence from each sample was normalized against an appropriate control sample that had not been incubated with hexane.

**Lapatinib stability assay**. Complexes were prepared at OP:SDS:CS:lapatinib ratios of 1:600:200:10 at a final concentration of 0.5 μM OP in PBS buffer. Slide-A-Lyzer dialysis cassettes (0.1–0.5 mL, 7000 Da MWCO, ThermoFisher Scientific, USA) were used to dialyze 500 μL of OP solution against 200 mL of PBS buffer containing 4 mg mL$^{-1}$ bovine serum albumin, a concentration equivalent to 10% fetal bovine serum. Dialysis was carried out at room temperature and aliquots were taken at 24, 48, and 72 h. Aliquots were diluted 8-fold in PBS for fluorescence spectroscopy, and lapatinib was detected by measuring emission spectra from 400 to 600 nm, with an excitation wavelength of 370 nm, as shown in Supplementary Fig. S14. For the native gel shown in Supplementary Fig. 15, OP:SDS:CS:lapatinib samples were prepared at a final concentration of 1.0 μM OP in PBS buffer. BSA:laptinib samples were prepared similarly. For the competition assays, equimolar amounts of BSA or OP:SDS:CS complexes were added to pre-formed lapatinib-protein complexes and incubated for 2 h at room temperature before analysis by agarose gel. Lapatinib fluorescence was detected by UV transillumination and protein was visualized with Coomassie blue.

**Cell viability assay**. Cytotoxicity was assessed using the WST-8-based Cell Counting Kit-8 from Sigma according to the manufacturer's instructions. HeLa cells were seeded at a density of 5000 cells per well in a 96-well plate in 100 μL of culture medium and allowed to recover at 37 °C and 5% $CO_2$ for 24 h. Protein, surfactant, and drug samples were prepared by serial dilution in sterile PBS. In every case, the input ratio of OP:SDS:CS:lapatanib was 1:600:200:10, which was used without further purification. A total volume of 25 μL sample was added to each well to provide the final concentrations shown in Fig. 5c, d. As a negative control, 10 μL of 10% Triton X-100 in PBS was used per well. The positive controls were just PBS. Cells were

incubated for 24 h in 5% $CO_2$ at 37 °C before the addition of 10 μL of CKK-8 reagent to each well. The plate was then incubated for 2–4 h at 37 °C and 5% $CO_2$ before measuring absorbance at 450 nm. The absorbance of CKK-8 in culture media, without cells, was used for background subtraction, and samples were normalized to untreated cells to provide the values shown in Fig. 5c, d. Samples were measured in sextuplicate with biological replicates of different protein batches.

**Statistics and reproducibility**. Experimental findings were reproduced on multiple occasions using both technical and biological replicates, all data were reproduced successfully. Specifically, native gel experiments (Figs. 2b, f, 5a and Supplementary Figs. 1c, e, 2, 10b, 15) were performed between three and ten times with similar results. Transmission electron microscopy (Fig. 2d, e and Supplementary Figs. 3, 11) was carried out three or four times with similar results in each case. Confocal microscopy (Fig. 4b, c and Supplementary Fig. 9) was carried out three to five times with similar results.

**Reporting summary**. Further information on research design is available in the Nature Research Reporting Summary linked to this article.

## Data availability
The data supporting the conclusions of this work can be found in the figures and Supplementary Information. Additional data are available from the corresponding author upon request. Macromolecular structural data from cryo-electron microscopy have been deposited in the EMDB with the accession codes: OP, EMD-10723; OP:SDS, EMD-10724; OP:SDS:CS, EMD-10725. Source data are provided with this paper.

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

## Acknowledgements

We would like to thank the Scientific Center for Optical and Electron Microscopy (ScopeM), ETH Zurich for help with TEM, cryo-EM, and CFM experiments and the Flow Cytometry Core Facility, ETH Zurich for technical support. We are grateful to the group of Prof. Nenad Ban for helpful discussions. We thank the research groups of Profs. P. Arosio and M. Kovalenko (D-CHAB, ETH) for the use of their Zetasizer instruments, and the group of Prof. R. Glockshuber (D-BIOL, ETH) for the use of their fluorimeter. This work was supported by the ETH Zurich and the European Research Council (Advanced ERC Grant ERC-AdG-2012-321295 to D.H.). T.E. is very grateful to the Human Frontier Science Program for a Long-term fellowship.

## Author contributions

T.G.W.E. and D.H. designed the project. T.G.W.E. conducted the experiments. S.T. obtained and analyzed electron microscopy data. All of the authors have agreed to the content of the manuscript.

## Competing interests

The authors T.G.W.E. and D.H. declare the following competing interests, a patent application "Edwardson & Hilvert, *Nanoparticles Encapsulating Small Molecules*, EP20157250.0" based on the OP:surfactant complexes described in this manuscript was filed by the ETH Zurich on the 13th of February, 2020. The author S.T. declares no competing interests.
