## [Peer Review File · Nature Communications]

REVIEWER COMMENTS

Reviewer #1 (Remarks to the Author):

The manuscript by T. Edwardson et al. reports SDS micelles loaded into a de novo designed protein cage. The templated micelles are further loaded with fluorescent dyes and hydrophobic drug molecules and their delivery is evaluated in a cell culture model. The experimental work is complete and of high quality. The results are clearly communicated and figures are excellent.

The approach is not entirely novel as functional micelles loaded to protein cages have already been presented (<https://doi.org/10.1016/j.jinorgbio.2014.01.004>) and the cage used here has been reported before. However, I'm not aware of such a complete and detailed protein cage loading study, which warrants the publication of the current manuscript.

Detailed remarks:

- Page 3: "This unique porous structure". The porous nature of the cage is not unique in my opinion. There are other protein cages with similar dimensions and pores e.g. ferritin from the Hyperthermophilic Archaeon *Archaeoglobus fulgidus* <https://doi.org/10.1016/j.str.2005.01.019>
- Page 4: "favourable hydrogen bond formation between the sulfate headgroup and the many arginine residues" Why hydrogen bond formation? Why is the main attraction not electrostatic?
- Page 4 / Figure 2b "bands with slightly increased mobility, indicating exposure of sulfate groups that increase surface charge" This claim should be verified with zeta potential measurements.
- Why are the surfactant molecules not observed in the cryo-TEM reconstructions? 800 molecules is a rather high number for such small space and I would assume that some density could be observed.
- How fast is the release in different conditions? Kinetics should be established through dialysis etc. experiments. Release in plain buffer seems to be rather slow, but in the cell experiments (Figure 4) release is stated to be more pronounced.
- Page S2: "purified by Ni-affinity and size-exclusion chromatography as previously reported" Reference is missing.
- DLS: The correlation function, PDI values and measurement angle should be reported.

Reviewer #2 (Remarks to the Author):

The manuscript by Edwardson et al. describes a novel method to create a hydrophobic core inside a porous protein cage. Inspired by composite compartments in nature, a designer protein cage with a positively charged lumen, OP, was used as a starting point for this work.

The literature is growing on novel protein cage designs, approaches to assembly and hierarchical structures using protein cages to generate novel materials and delivery systems. This manuscript is an excellent addition to this growing field. It is, as far as I know, completely novel in its use of small molecule surfactants to generate a hydrophobic core within a protein cage and that carries important implications for processing and generalisability to other small protein cages.

Furthermore, the lipid composition of the encapsulated micelle/complex can be tuned and the loading of hydrophobic compounds is demonstrated. Both the model fluorescent Nile Red and active pharmaceutical ingredient lapatinib are shown with efficient delivery to the cytosol of

mammalian cells. It is noted that the OP cage is effectively taken up by numerous cell types.

Because of the novelty of the approach, quality of the results and associated experimental design, and the implications for expanding the design space for novel protein-based assemblies, I think this manuscript will make an important contribution to the literature and will be of broad interest. The style of communication is very clear and overall makes for a compelling manuscript.

The results show thoughtful and rigorous experimentation and are high quality throughout. Extensive characterisation using both ensemble and single particle methods show that the cage structure is maintained with and without encapsulated micelles of various composition. Orthogonal methods determine an SDS capacity of ~ 800 molecules per OP cage. My only comment on this interpretation is that there is the possibility that SDS is also associated with the exterior of the protein cage via the usual hydrophobic interactions. It is not possible to tell from the cryo-EM structures. It is, however, something that could be investigated by zeta-potential measurements; perhaps the authors can comment on the nature of the outer surface of the cage that leads them to conclude that the surfactant is only on the inside? Does the theoretical capacity of the OP cage lumen take the geometry of a phase separated micelle into account?

The author speculation on cholesterol sulfate stabilising the encapsulation complex brings up an important discussion point; the stability of the complex and lipophilic compound retention. It would be helpful to have an idea of the magnitude of this effect (that is, versus SDS alone) and some comments on whether this is likely to be compound-specific or even enhanced by the lipophilic compound itself. Is there data on Nile Red retention aside from the flow cytometry data, which seems to suggest that retention is relatively low with SDS alone?

This approach works well with the relative small OP protein cage and it is generalizable to some extent. However, micelle stability likely depends on the combination of electrostatic interactions with the cage interior surface and enhancement of hydrophobic interactions between surfactant tails due to the confines of the cage. Would this approach lead to stable micelle complex formation inside larger protein cages following charge-driven loading? Previous attempts to impart hydrophobic cores on soluble protein cages have used constrained chemical polymerisation within virus-derived protein cages have effectively created composite particles. These similar outcomes from very different approaches would help frame the novelty here.

Reviewer #3 (Remarks to the Author):

This is an elegant and simple idea of modulating the interior properties of a protein cage to accommodate different types of molecules. Electrostatic interactions between the interior surface and surfactant headgroups were used to drive encapsulate a surfactant micelle which could subsequently be used to drive solubilization of small hydrophobic molecules within the cage.

There are a few aspects to the manuscript that require some further explanation/clarification but overall this is a nice piece of work.

1. The authors state that there are 144 positively charged residues on the inside of the capsid, however 800 surfactant molecules are incorporated creating a serious charge imbalance. Is there an ionic strength dependence for the incorporation? What is the charge on the exterior of the OP capsid and is there a change in the zeta potential upon SDS loading?
2. Is there evidence for dynamic exchange between SDS molecules inside OPs (or even a bulk micelle)? Experiments were all done below the CMC, does the same encapsulation take place above the CMC?
3. Figure 2B shows that as the SDS concentration is increased there is an increased exposure of sulfate groups. Does that mean that the capsid is falling apart?

4. Figure 2C has a few peaks that appear in the baseline region of the chromatogram containing OP+SDS. What are they? Are the capsids falling apart? Is there a drop in OP concentration after exposure to SDS?
5. Figure 3D: Can an inset be provided for the figure that illustrates the red shift in the third stage?
6. It is not clear whether the ratio, OP:SDS:CS:Nile Red = 1:600:200:5, is input or output. If it is output, there is no data to support it. (In the cell viability experiments too, it is not clear whether the ratio, OP:SDS:CS:lapatinib = 1:600:200:10, is input or output. If it is input, how was the total concentration of lapatinib quantified?)
7. When the authors state that the dye escaped the capsids inside the cell. Why isn't there a decrease in fluorescence intensity or does the molecule partition into somewhere in the cell that provides a similar hydrophobic environment as the micelle within the capsid?
8. How was the incorporation of CS verified? Please explain Figure S8D in more detail.

Minor

1. The introduction, page 2, states that "the incorporation of molecules other than proteins and nucleic acids has not been extensively explored", however there has been plenty of research encapsulating polymers, metal ions, etc into protein cages. Those works should be cited and the claim toned down a bit.
2. In Figure 2F: Are the concentrations of ssDNA drastically different between the two samples? Where is the excess ssDNA?
3. The structures in S4 and S5 appear different. Is SDS making a contribution to these differences and can this be quantified and explained?
4. No aggregates were observed in TEM of OP-SDS cages (Figure S3). However, Cryo-EM on same sample (Figure S5-a) showed presence of larger size aggregates. Similar larger size aggregates were also observed in Cryo-Em of OP-SDS-CS cages. Could this be due to degradation of OP-SDS & OP-SDS-CS cages?

Reviewer #1 (Remarks to the Author):

The manuscript by T. Edwardson et al. reports SDS micelles loaded into a de novo designed protein cage. The templated micelles are further loaded with fluorescent dyes and hydrophobic drug molecules and their delivery is evaluated in a cell culture model. The experimental work is complete and of high quality. The results are clearly communicated and figures are excellent.

The approach is not entirely novel as functional micelles loaded to protein cages have already been presented (<https://doi.org/10.1016/j.jinorgbio.2014.01.004>) and the cage used here has been reported before. However, I'm not aware of such a complete and detailed protein cage loading study, which warrants the publication of the current manuscript.

As discussed in the main text (page 6, paragraph 2), an important distinction between our work and previously reported systems which use viral capsid proteins to coat pre-formed micelles or nanospheres, is that the protein cage acts to template the formation of micelles from dispersed amphiphiles. As such, our design concept differs from the cited examples, all of which exploit pH-inducible assembly/disassembly of the CCMV protein to either encapsulate or coat existing particles. We thank the reviewer for the suggested reference (27: Millán et al. *J. Inorg. Biochem.* **2014**), which reports another example of capsid protein assembly around multiple micelles. We have added this citation to the relevant discussion on page 6.

Detailed remarks:

- Page 3: "This unique porous structure". The porous nature of the cage is not unique in my opinion. There are other protein cages with similar dimensions and pores e.g. ferritin from the Hyperthermophilic Archaeon *Archaeoglobus fulgidus* <https://doi.org/10.1016/j.str.2005.01.019>

We agree that porosity is not unique to the OP protein cage. However, to our knowledge, there are no other examples of porous protein cages with octahedral symmetry, similar dimensions, and positively charged lumen. To clarify though, we have modified our original statement, noting that this "*highly stable, porous structure*", together with the positive charge, are the key features of the cage used in this study.

- Page 4: "favourable hydrogen bond formation between the sulfate headgroup and the many arginine residues" Why hydrogen bond formation? Why is the main attraction not electrostatic?

Yes, although the sulfate-guanidinium pair is well-matched for hydrogen bonding, the main, and longer ranging, non-covalent interaction is indeed electrostatic. This has been confirmed experimentally by fluorescence measurements for OP:SDS complex formation, using Nile Red as the probe for micelle formation. These data (Supplementary Fig. S6) show that the rate of assembly is ionic strength dependent, with faster assembly at lower [NaCl], consistent with an electrostatically driven self-assembly. We have also reworded the sentence on page 4 to read "*favourable electrostatic interactions and hydrogen bond formation*" and added a reference for the salt bridge character of the arginine-sulfate interaction (22: Rozas & Kruger

J. Chem. Theory Comput. **2005**).

- Page 4 / Figure 2b “bands with slightly increased mobility, indicating exposure of sulfate groups that increase surface charge” This claim should be verified with zeta potential measurements.

As suggested, we have carried out zeta potential measurements on the empty OP cage and the most relevant OP:SDS complex containing 800 equivalents of SDS (Supplementary Fig. S1d). The data indicate negligible change in the surface charge of the protein cage upon internalization of the SDS molecules, which is consistent with the native gel analysis in Figs. 2b and S1b. Nevertheless, to obtain these data we had to prepare samples in pure water to obtain a measurable signal and avoid destruction of the cuvette electrodes by the normal buffers. The samples also had to be measured at relatively high concentrations (48 μ M protein), which prevented reliable measurement of the OP:SDS complexes at higher ratios of surfactant. This is likely due to SDS micellization at these high concentrations, which are in the range of the CMC.

As an alternative means to deconvolute the gel electrophoresis results and determine whether the change in electrophoretic mobility is due to an increase in charge or a decrease in size, we carried out DLS measurements on OP with increasing equivalents of SDS (Supplementary Fig. S1a,b). These measurements could be carried out in PBS buffer at the same concentrations used for the gel experiment and revealed that the diameter of the particles remained constant in the presence of 200 to 2000 equivalents of SDS. As such, it can be concluded that the change in electrophoretic mobility seen in Fig. 2b is due to an increase in surface charge associated with incompletely encapsulated lipid at the highest SDS concentrations tested. Although the exact localization of the additional SDS molecules is as yet unknown, it is likely that they partially fill the pores of the protein cage, exposing their negatively charged headgroups. We would also like to point out that more than 800 equivalents exceeds the optimal ratio we determined for the creation of stable protein-micelle complexes. The main purpose of these experiments was to demonstrate that there is an optimal loading capacity, which is now corroborated by theoretical calculations, gel electrophoresis, fluorescence measurements, DLS and zeta potential.

- Why are the surfactant molecules not observed in the cryo-TEM reconstructions? 800 molecules is a rather high number for such small space and I would assume that some density could be observed.

What is seen in these 3D representations of the cryo-EM maps depends on data quality, order, symmetry and display thresholds.

Here, based on the known structure of the OP cage we imposed octahedral symmetry during reconstruction, which smears out any density that does not have the same symmetry. Thus, what could be expected is a cloud of density of lower threshold, similar to what is seen for the nucleic acid genomes in the EM structures of virus particles. Examples of a medium-resolution viral structure can be found in EMD entries EMD-1886 or EMD-5268, which show different viral capsids with fuzzy and significantly weaker densities for membranes and genomes.

That said, there are clear indications for the presence of surfactant molecules in the well behaved OP:SDS:CS sample. To illustrate this point, we added a central slice through the reconstruction for all datasets (Supplementary Figs. S4, S5 and S12). The central slice representation for OP:SDS:CS (Supplementary Fig. S12f) reveals a diffuse density within the capsid cavity. As in the case of viral RNA, the density is lower than that of the symmetric protein shell. For the 3D volumes shown in Supplementary Fig. S12, the display threshold was chosen such that the cage structure is well depicted. Lowering the threshold would increase the luminal density, but also blur out the protein features.

The OP:SDS sample (Supplementary Fig. S5) was the most difficult to work with, and did not respond well to being concentrated. Nevertheless, multiple datasets were collected and all showed that the quaternary cage structure was intact, although the resolution and cavity density varied. Some datasets did show some interior density, but we opted to use the best resolved reconstruction as an objective criterion for the final datasets of all samples.

The data presented here were collected on a 200 kV microscope and relied on about 100 images per reconstruction, which was sufficient to confirm cage integrity. More detailed analyses would require more data, and would benefit from the higher signal-to-noise ratios provided by a high-end microscope. Nonetheless, we did not consider this necessary as the crystal structure of the cage is already known and the various other biophysical experiments convincingly define the nature of the capsid:cargo complex.

In summary, the EM data presented confirm that the capsid structure remains intact in the presence of high local surfactant concentration. Although they are also consistent with internalization of the lipids, the poorly resolved luminal densities should not be overinterpreted.

- How fast is the release in different conditions? Kinetics should be established through dialysis etc. experiments. Release in plain buffer seems to be rather slow, but in the cell experiments (Figure 4) release is stated to be more pronounced.

In addition to dialysis and gel electrophoresis experiments with lapatinib loaded in OP:SDS:CS against albumin-containing buffer, we have carried out a direct comparison of Nile Red release kinetics for OP:SDS and OP:SDS:CS (Supplementary Fig. S13). These new data highlight the relatively slow rate of release and confirm that addition of cholesterol sulfate to the surfactant mixture increases complex stability. Additionally, the timescale for cargo release (45-65% at 16 hours) is consistent with the cell studies, where cells were analyzed 16- 24 hours after treatment with protein cages. An important additional consideration for cargo release within cells is the presence of many lipid membranes, which provide a highly favourable environment for the membrane probe Nile Red and may potentially disrupt the protein-surfactant complexes leading to faster cargo release. We have added a sentence to the discussion of the cell experiments (Page 9, line 1) that explains the release of Nile Red from the capsids in the context of competing intracellular environments.

- Page S2: “purified by Ni-affinity and size-exclusion chromatography as previously reported” Reference is missing.

We have added the appropriate reference to Supplementary Information page S2.

- DLS: The correlation function, PDI values and measurement angle should be reported.

We have added the correlation functions to Supplementary Figs. S1 and S10 and the average diameters and PDI values to the figure captions. These details have also been added for the new DLS data added to Supplementary Fig. S1. The measurement angle of 173° has been added to the methods section (page S3).

Reviewer #2 (Remarks to the Author):

The manuscript by Edwardson et al. describes a novel method to create a hydrophobic core inside a porous protein cage. Inspired by composite compartments in nature, a designer protein cage with a positively charged lumen, OP, was used as a starting point for this work.

The literature is growing on novel protein cage designs, approaches to assembly and hierarchical structures using protein cages to generate novel materials and delivery systems. This manuscript is an excellent addition to this growing field. It is, as far as I know, completely novel in its use of small molecule surfactants to generate a hydrophobic core within a protein cage and that carries important implications for processing and generalisability to other small protein cages. Furthermore, the lipid composition of the encapsulated micelle/complex can be tuned and the loading of hydrophobic compounds is demonstrated. Both the model fluorescent Nile Red and active pharmaceutical ingredient lapatinib are shown with efficient delivery to the cytosol of mammalian cells. It is noted that the OP cage is effectively taken up by numerous cell types.

Because of the novelty of the approach, quality of the results and associated experimental design, and the implications for expanding the design space for novel protein-based assemblies, I think this manuscript will make an important contribution to the literature and will be of broad interest. The style of communication is very clear and overall makes for a compelling manuscript.

The results show thoughtful and rigorous experimentation and are high quality throughout. Extensive characterisation using both ensemble and single particle methods show that the cage structure is maintained with and without encapsulated micelles of various composition. Orthogonal methods determine an SDS capacity of ~800 molecules per OP cage. My only comment on this interpretation is that there is the possibility that SDS is also associated with the exterior of the protein cage via the usual hydrophobic interactions. It is not possible to tell from the cryo-EM structures. It is, however, something that could be investigated by zeta-potential measurements; perhaps the authors can comment on the nature of the outer surface of the cage that leads them to conclude that the surfactant is only on the inside? Does the theoretical capacity of the OP cage lumen take the geometry of a phase separated micelle into account?

The location of the SDS molecules relative to the protein cage is indeed a crucial aspect in the characterization of these complexes. Based on the data collected, we can confidently conclude that in the presence of up to 800 molecules per capsid the surfactant molecules are fully encapsulated within the lumen. As suggested, and as noted in our response to reviewer #1, we have now carried out zeta potential measurements on the empty OP cage and the OP:SDS complex with 800 SDS molecules per capsid. These measurements reveal negligible change in particle charge upon internalization of SDS molecules. These data corroborate the gel electrophoresis data, and other structural analyses that show no change in electrophoretic mobility or capsid size, which is only possible if the SDS molecules are encapsulated within the lumen as intended. As discussed on page 6 and detailed in the Supplementary information, our luminal capacity calculation takes into account the volume occupied by SDS molecules in a phase separated micelle (Hammouda, *J. Res. Natl. Inst. Stand. Technol.* **2013**, 118, 151). We therefore conclude that *“Although the exact internal organization of the SDS aggregates is unknown, the packing density suggests that the molecules are arranged with some structural similarity to their typical oblate ellipsoid micellar form.”*

The author speculation on cholesterol sulfate stabilising the encapsulation complex brings up an important discussion point; the stability of the complex and lipophilic compound retention. It would be helpful to have an idea of the magnitude of this effect (that is, versus SDS alone) and some comments on whether this is likely to be compound-specific or even enhanced by the lipophilic compound itself. Is there data on Nile Red retention aside from the flow cytometry data, which seems to suggest that retention is relatively low with SDS alone?

The ability to tune loading and release kinetics by altering the lipid composition is an important feature of this encapsulation system and we appreciate the reviewer’s suggestion to determine this more directly. We have now carried out a direct comparison of Nile Red retention in OP:SDS and OP:SDS:CS complexes (Supplementary Fig. S13). As the complexes are quite stable and release kinetics are slow, we developed a dual-phase competition assay using an organic solvent as a sink for Nile Red molecules released from the protein complexes. The data corroborate the flow cytometry data and demonstrate that the addition of CS stabilizes the complex considerably. After 16 hours, the OP:SDS:CS complexes retain 55% of their Nile Red cargo, which is 20% more than the OP:SDS complexes.

This approach works well with the relative small OP protein cage and it is generalizable to some extent. However, micelle stability likely depends on the combination of electrostatic interactions with the cage interior surface and enhancement of hydrophobic interactions between surfactant tails due to the confines of the cage. Would this approach lead to stable micelle complex formation inside larger protein cages following charge-driven loading? Previous attempts to impart hydrophobic cores on soluble protein cages have used constrained chemical polymerisation within virus-derived protein cages have effectively created composite particles. These similar outcomes from very different approaches would help frame the novelty here.

As mentioned in the manuscript, the size and structure of the OP protein cage is crucial for the formation of the unique complexes that we have studied. Nevertheless, as we

speculate in the conclusion, the concept of combining orthogonal electrostatic and hydrophobic interactions to create composite systems is in principle applicable to a range of cages. However, it is indeed likely that different morphologies will be formed and there may be different requirements for the amphiphilic components when larger cages are used. With this in mind we have augmented the concluding discussion to more directly address these possibilities and help frame the novelty of our self-assembly concept, also in the context of protein cages filled with polymers generated by in situ polymerization.

Reviewer #3 (Remarks to the Author):

This is an elegant and simple idea of modulating the interior properties of a protein cage to accommodate different types of molecules. Electrostatic interactions between the interior surface and surfactant headgroups were used to drive encapsulation of a surfactant micelle which could subsequently be used to drive solubilization of small hydrophobic molecules within the cage.

There are a few aspects to the manuscript that require some further explanation/clarification but overall this is a nice piece of work.

1. The authors state that there are 144 positively charged residues on the inside of the capsid, however 800 surfactant molecules are incorporated creating a serious charge imbalance. Is there an ionic strength dependence for the incorporation? What is the charge on the exterior of the OP capsid and is there a change in the zeta potential upon SDS loading?

The OP protein cage was engineered by addition of 144 arginines to the interior surface of the parent cage, O3-33. Although there are additional positively charged residues, e.g. Lys41 and Lys135, present on the luminal surface, the internalization of 800 monovalent anions still leads to a charge imbalance. The interplay of counterions, other amino acid residues, hydrophobic interactions between surfactant molecules and entropic penalties for free surfactant molecules are all responsible for the formation of these complexes. In all of our previous experiments, counterions were present at ionic strengths in the range of 0.15-0.23 mol/L. We have now added kinetic data obtained from fluorescence assays (Supplementary Fig. S6), which show that complex formation is favourable across a range of ionic strengths ($I = 0.01-0.15$ mol/L). The data also reveal an inverse relationship between ionic strength and assembly rate, confirming that complex formation is an electrostatically driven process.

As expected from the protein structure and demonstrated by gel electrophoresis, the exterior of the OP cage is negatively charged at pH >7. We have now carried out zeta potential measurements (Supplementary Fig. S1d) on the empty and loaded cage, which show negligible change in zeta potential upon internalization of 800 surfactant molecules.

2. Is there evidence for dynamic exchange between SDS molecules inside OPs (or even a bulk micelle)? Experiments were all done below the CMC, does the same encapsulation take place above the CMC?

We have not specifically investigated this aspect of the complexes. However, we expect that release of lipids from the capsids is slower than that observed for Nile Red or lapatinib, due to the favourable electrostatic interactions between the capsid surface and both SDS and cholesterol sulfate. Furthermore, the mechanisms of exchange between micelles often involve fission, fusion or collision (Rharbi et al. *J. Am. Chem. Soc.* **2004**, 126, 6025; Montigny et al. *PLoS One* **2017** 12, e0170481), processes that are likely hindered by the outer protein shell.

Above the CMC, the mechanism of complex formation will be quite different. However, it could be expected that the final complexes would be similar in structure at the same ratios of surfactant to protein. To test this hypothesis, we have carried out native gel analysis of OP:SDS complexes assembled at increasing SDS concentrations, up to the reported CMC in PBS, (Supplementary Fig. S1e). The data show that there is no change in complex structure with increasing SDS concentration, suggesting that the same complexes are formed under these conditions.

3. Figure 2B shows that as the SDS concentration is increased there is an increased exposure of sulfate groups. Does that mean that the capsid is falling apart?

As shown in Supplementary Fig. S1c, at very high SDS concentrations (150-fold higher than shown in Figure 2b) the capsid does indeed fall apart. However, within the concentration ranges shown in Figure 2b, the capsids remain intact. We have now added a DLS comparison of capsid radii between 0 and 2000 equivalents of SDS (Supplementary Fig. S1a), which confirms that there is negligible change in size, showing that the capsids do not fall apart and the change in electrophoretic mobility is due to an increase in negative charge. We conjecture that this increase in negative charge may be due to additional SDS molecules filling the pores of the protein cage, which would increase surface charge without changing the overall diameter. Importantly, at the optimal loading capacity that we have used for all of the other experiments (800 molecules of surfactant), we can be confident that the SDS molecules are fully localized within the interior cavity of the capsid based on the native gel, dynamic light scattering, zeta potential, size-exclusion chromatography and electron microscopy data.

4. Figure 2C has a few peaks that appear in the baseline region of the chromatogram containing OP+SDS. What are they? Are the capsids falling apart? Is there a drop in OP concentration after exposure to SDS?

The minor peaks seen in Figure 2c can be attributed to two things. Firstly, small RNA fragments can be present in the OP capsid after purification. Due to the positively charged nature of the protein cage, RNA from the *E. coli* expression host is encapsulated during production. The vast majority of these contaminants are removed during purification, but some batches retain a fraction of RNA fragments. While these molecules are tightly bound to the interior of the cage in the OP sample, they are displaced by SDS internalization, and are seen as late eluting peaks in the SEC traces. Secondly, there is a slight difference between the sample buffer composition and the eluent, which was used to zero the baseline. As such these molecules will cause a change in absorbance as they elute from the column. This is most notable in the OP:SDS:CS sample (Supplementary Fig. S10a), which contains DMSO from the cholesterol sulfate solution. As there is no drop in OP concentration after exposure to

800 equivalents of SDS, the absence of any notable peaks in SEC or changes in electrophoretic mobility in the gel experiments are in line with the other structural data showing that the complexes do not fall apart under these conditions.

5. Figure 3D: Can an inset be provided for the figure that illustrates the red shift in the third stage?

We have added these spectra to the supplementary information (Supplementary Fig. S7) as the main figure is already quite busy.

6. It is not clear whether the ratio, OP:SDS:CS:Nile Red = 1:600:200:5, is input or output. If it is output, there is no data to support it. (In the cell viability experiments too, it is not clear whether the ratio, OP:SDS:CS:lapatinib = 1:600:200:10, is input or output. If it is input, how was the total concentration of lapatinib quantified?)

All of these ratios represent the input. The samples were prepared at concentrations and stoichiometries that were determined to give quantitative complex formation so that no additional purification or quantification steps were needed. We have added details to the relevant methods sections (pages S5, S6 and S7).

7. When the authors state that the dye escaped the capsids inside the cell. Why isn't there a decrease in fluorescence intensity or does the molecule partition into somewhere in the cell that provides a similar hydrophobic environment as the micelle within the capsid?

Yes, partitioning of the dye into the many hydrophobic environments present within a cell is expected and provides a reasonable explanation for the fluorescence observed in the cell experiments. We have modified a sentence in the cell study discussion (page 9, line 1) to clarify this point.

8. How was the incorporation of CS verified? Please explain Figure S8D in more detail.

As the fluorescence emission of Nile Red is highly sensitive to its local environment, it was used as a probe to show CS incorporation within OP:SDS complexes. We have slightly expanded the caption for Supplementary Fig. S10e to make this clearer. We also note that the incorporation of CS within OP:SDS:CS complexes can be inferred from the difference in release kinetics observed in comparison to OP:SDS (Supplementary Fig. S13) and cellular uptake (Fig. 4d).

Minor

1. The introduction, page 2, states that “the incorporation of molecules other than proteins and nucleic acids has not been extensively explored”, however there has been plenty of research encapsulating polymers, metal ions, etc into protein cages. Those works should be cited and the claim toned down a bit.

While a range of molecules have been encapsulated within protein cages, we find that hierarchical incorporation of lipid molecules to create a new luminal phase, is both conceptually and practically different. Incorporating lipids in the design of hybrid protein cage materials, rather than encapsulation of other materials, such as metal

ions, nanoparticles and polymers, is also more relevant in the context of biological assemblies. To clarify this aspect, we have changed the cited sentence to specify 'biological molecules' in an attempt to clearly put this in the context of the hybrid systems seen in nature. We have also modified the conclusion to compare and contrast our approach with the encapsulation of polymers, which is most relevant as it provides an alternative means to create hydrophobic cores within protein cages.

2. In Figure 2F: Are the concentrations of ssDNA drastically different between the two samples? Where is the excess ssDNA?

The DNA concentration is actually exactly the same in both samples. The difference in fluorescence is due to quenching of the DNA-conjugated fluorophore upon encapsulation in the OP protein cage, as previously reported (Edwardson et al. *J. Am. Chem. Soc.* 2018, 140, 10439). At the indicated concentration and stoichiometry, the encapsulation of DNA is quantitative, hence no excess DNA is to be expected. As these details were evidently unclear, we now explain them in the relevant methods section (page S3).

3. The structures in S4 and S5 appear different. Is SDS making a contribution to these differences and can this be quantified and explained?

The difference between the OP (Supplementary Fig. S4) and OP:SDS (Supplementary Fig. S5) structures is mainly one of data quality, particle number, and consequently resolution. The lower resolution of the latter creates smoother features, which explains the observed difference in the 3D density representation. Despite extensive efforts to optimize grid preparation and freezing conditions, the OP:SDS samples were not well behaved, making cryo-EM analysis difficult. This was not the case for OP:SDS:CS, which afforded similar resolution to the OP control, explaining why alpha-helical features are observed. Nevertheless, it is clear from the data obtained that the protein structure is intact and unperturbed in all cases.

4. No aggregates were observed in TEM of OP-SDS cages (Figure S3). However, Cryo-EM on same sample (Figure S5-a) showed presence of larger size aggregates. Similar larger size aggregates were also observed in Cryo-Em of OP-SDS-CS cages. Could this be due to degradation of OP-SDS & OP-SDS-CS cages?

The large, darker structures seen in the cryoEM micrograph in Supplementary Fig. S5a are not protein aggregates, but contaminants in the liquid ethane used for freezing the samples (Grassucci et al., *Nat. Protoc.* **2007**, 3239–3246). Due to particle picking, manual inspection, and 2D- and 3D-classification, these do not contribute to the final reconstructions.

REVIEWERS' COMMENTS:

Reviewer #1 (Remarks to the Author):

Authors have addressed all my comments appropriately.

Reviewer #2 (Remarks to the Author):

The authors have compiled thorough and considered responses to all referee comments. They have provided extensive explanation and appropriate experimentation where necessary. The end result is a detailed and high quality report into a novel biocompatible templating strategy for the assembly of hybrid protein cages with hydrophobic lumen.

Reviewer #3 (Remarks to the Author):

The authors have adequately addressed my scientific concerns in a substantive way. I think this is an innovative piece of work that deserves the recognition of publication as a Nature Comm paper.

I am disappointed that there is not a recognition reflected in the literature citations of those who did some of the original and seminal work in this field.

REVIEWERS' COMMENTS:

Reviewer #1 (Remarks to the Author):

Authors have addressed all my comments appropriately.

We are glad that the reviewer is satisfied with our revised manuscript.

Reviewer #2 (Remarks to the Author):

The authors have compiled thorough and considered responses to all referee comments. They have provided extensive explanation and appropriate experimentation where necessary. The end result is a detailed and high quality report into a novel biocompatible templating strategy for the assembly of hybrid protein cages with hydrophobic lumen.

We appreciate this positive feedback.

Reviewer #3 (Remarks to the Author):

The authors have adequately addressed my scientific concerns in a substantive way. I think this is an innovative piece of work that deserves the recognition of publication as a Nature Comm paper.

We are glad that the reviewer is satisfied with our revised manuscript.

I am disappointed that there is not a recognition reflected in the literature citations of those who did some of the original and seminal work in this field.

While we would gladly alter our bibliography to include any relevant research that we might have missed, without concrete examples we are at a loss to identify which work may have been omitted. Nevertheless, we have added primary citations for two pioneering examples of combining protein cages with inorganic nanoparticles and polymers that were previously covered in review articles (Refs. 5, 9, 10, 11 and 38). With regards to work that is more closely related to our research, we believe that we have thoroughly and appropriately cited all of the relevant literature to put our study in its proper context within the field of protein cage engineering. Our referencing of hybrid systems that comprise protein cages and lipids is exhaustive, citing original articles in each case. Protein cage studies that are not directly related have been recognised by the inclusion of review articles from the prominent researchers in the field.